# ProtoNMF: Turning a Black Box into a Prototype Based Interpretable Model via Non-Negative Matrix Factorization

## Abstract

Models using parts of images as prototypes for interpretable image classification are receiving increasing attention due to their abilities to provide a transparent reasoning process in a "this looks like that" manner. However, existing models are typically constructed by incorporating an additional prototype layer before the final classification head, which often involve complex multi-stage training procedures and intricate loss designs while under-performing their black box counterparts in terms of accuracy. In order to guarantee the recognition performance, we take the first step to explore the reverse direction and investigate how to turn a trained black box model into the form of a prototype based model. To this end, we propose to leverage the Non-negative Matrix Factorization (NMF) to discover interpretable prototypes due to its capability of yielding parts based representations. Then we use these prototypes as the basis to reconstruct the trained black box's classification head via linear convex optimization for transparent reasoning. Denote the reconstruction difference as the residual prototype, all discovered prototypes together guarantee a precise final reconstruction. To the best of our knowledge, this is the first prototype based model that guarantees the recognition performance on par with black boxes for interpretable image classification. We demonstrate that our simple strategy can easily turn a trained black box into a prototype based model while discovering meaningful prototypes in various benchmark datasets and networks.

## 1 Introduction

In the past years, deep learning based models such as Convolutional Neural Networks (CNN) and Vision Transformer (ViT) have achieved impressive performance in computer vision tasks (He et al., 2016; Dosovitskiy et al., 2020; Girshick, 2015; Ronneberger et al., 2015). However, the interpretability of these models has always been a major concern and limits their real-world deployments.

Among many works trying to tackle this problem, the prototype based models are receiving increasing attention because they don't require additional interpretability supervision while guaranteeing a transparent reasoning process (Ukai et al., 2022). Moreover, the explanation offered in a "this looks like that" manner is easy for human to understand (Wang et al., 2021; Nauta et al., 2021; Rymarczyk et al., 2022; Donnelly et al., 2022; Rymarczyk et al., 2021; Chen et al., 2019; Ukai et al., 2022). The reasoning process includes the comparisons of extracted features with prototypes. The final decision is made via linear combination of feature similarities to different prototypes weighted by the coefficients, where the contributions of different components can be clearly identified. Therefore, these models are considered inherently interpretable as the interpretable decision making process is incorporated into the model's architecture design (Chen et al., 2019).

Existing prototype based models are typically constructed by additionally incorporating a prototype layer, requiring complex training scheme, intricate loss designs, yet typically under-performing their black box counterparts in terms of recognition accuracy. In this work, we try to explore the reverse direction and aim to address the following question: **can we turn a trained black box model into a prototype based model while discovering interpretable prototypes for transparent reasoning?**

Inspired by psychological (Biederman, 1987) and physiological (Wachsmuth et al., 1994) study that human recognize objects by components, we seek for discovering a set of prototypes which could indicate different parts of the target object for each class. In addition, we encourage the discovered prototypes to be diverse yet complete (We call it comprehensive for ease of description in later sections). Take bird species classification as an example, an intuitive understanding is that the discovered prototypes should indicate different body parts, yet sum together to cover the whole bird body. For the diversity: it is straightforward to see that duplicate prototypes are unnecessary. For the completeness: although incomplete prototypes may also be enough to discriminate one class from another, a set of more complete prototypes offers users more chance for further improvement of the model. For example, if the model wrongly relies on some discriminative but biased prototype (e.g., background or some wrong concepts), human may conduct test-time intervention (Koh et al., 2020) to manually suppress its contribution and increase the importance of the correct prototypes using expert knowledge. This won't be possible if the set of prototypes are less complete where the correct prototype may not even exist in the discovered set.

To this end, we propose to leverage the popular signal decomposition technique called Non-negative Matrix Factorization (NMF) (Lee & Seung, 1999) for the prototype discovery. Unlike principle component analysis, NMF is known to offer parts based representations (Wang & Zhang, 2012), making it suitable to serve as prototypes indicating parts of the target object. Moreover, the optimization process of NMF explicitly optimizes the prototypes to be comprehensive. In the context of prototype based models with transparent reasoning process, we propose to first apply NMF in the features extracted by a trained model for the prototype discovery. Then we try to reconstruct the black box's classification head via a linear optimization process using the discovered prototypes as the basis. We use the linear combination in order to follow the "this looks like that" framework (Chen et al., 2019) and enable the contribution of each component to be clearly identified. In the end, we calculate the reconstruction difference and take it as the residual prototype to guarantee the performance on par with the black box models. Our contributions are:

- We introduce ProtoNMF, the first prototype based model that guarantees the recognition performance on par with black boxes for interpretable image classification.

- We propose to leverage NMF for the prototype construction and use the discovered prototypes to build a transparent reasoning process. Since NMF offers parts based representations, they are suitable to indicate image parts as prototypes.

- We conduct extensive experiments to demonstrate that our simple strategy can easily turn a trained black box into a prototype based model while discovering meaningful prototypes in multiple benchmark datasets and network architectures.

## 2 RELATED WORKS

**Prototype based models** One prominent work designing prototype based models for interpretable image classification is the ProtopNet (Chen et al., 2019). This work does not focus on offering quantitatively better interpretability. Instead, it focuses on offering a transparent reasoning process. Given an input image, the model compares image parts with the prototypes, and make the predictions based on a weighted combination of the similarity scores between image parts (feature patches from the input's feature maps) and prototypes (one prototype is a specific feature patch from a specific feature map). This model provides inherent interpretability of the decision making in a "this looks like that" manner. Following this framework, many works are proposed to investigate different aspects, such as discovering the similarities of prototypes (Rymarczyk et al., 2021), making the prototype's class assignment differentiable (Rymarczyk et al., 2022), making the prototypes spatially flexible (Donnelly et al., 2022), combining it with decision trees (Nauta et al., 2021) or K-nearest neighbors (KNN) (Ukai et al., 2022). Although above methods have different motivations of improvement, their major evaluation metric is still the performance. Even though, none of these methods can guarantee the performance on par with the black box counterparts.

**Post-hoc methods** These methods focus on explaining trained black box models. Therefore, our model can be categorized to both a prototype based model and a post-hoc method. Prior post-hoc methods can roughly be categorized to perturbation (Ribeiro et al., 2016; Zeiler & Fergus, 2014; Zhou & Troyanskaya, 2015) based or backpropagation (Selvaraju et al., 2017; Zhang et al., 2018;

Sundararajan et al., 2017) based. These methods primarily emphasize local interpretability, providing explanations specific to individual inputs. However, they often fall short in terms of capturing global knowledge and utilizing it for transparent reasoning processes. In contrast, our method focuses on exploring the global knowledge for constructing a prototype based reasoning process in interpretable image classification. This process brings more insights via pointing out to what prototype certain areas look similar to, while some attention based methods such as Grad-CAM (Selvaraju et al., 2017) can only explain where the network is looking at (Rudin, 2019). Another related work in audio processing (Parekh et al., 2022) utilizes NMF in post-hoc explanations. However, this work is not designed for images and no transparent reasoning process is constructed.

**Matrix factorization and basis decomposition** It is not new to understand complex signals via the decomposition. Conventional methods such as NMF (Lee & Seung, 1999), Principle Component Analysis (Frey & Pimentel, 1978), Vector Quantization (Gray, 1984), Independent Component Analysis (Hyvärinen & Oja, 2000), Bilinear Models (Tenenbaum & Freeman, 1996) and Isomap (Tenenbaum et al., 2000) all discover meaningful subspace from the raw data matrix in an unsupervised manner (Zhou et al., 2018). Among these works, NMF (Lee & Seung, 1999) stands out as the only approach capable of decomposing whole images into parts based representations due to its use of non-negativity constraints which allow only additive, not subtractive combinations. In the era of deep learning, (Collins et al., 2018) finds that NMF is also effective for discovering interpretable concepts in CNN features with ReLU (Nair & Hinton, 2010) as activation functions. However, this work does not leverage the discovered concepts from the training data to construct a model with transparent reasoning process. Moreover, the usage of NMF is not explored in more recent architectures which allow negative feature values (e.g., ViT (Dosovitskiy et al., 2020) or CoC (Ma et al., 2023)). Another related work is (Zhou et al., 2018), which approximates the classification head vector via a set of interpretable basis vectors to facilitate interpretable decision making. However, this work leverages the concept annotations to obtain the interpretable basis rather than employing an unsupervised approach.

**Networks with additional interpreters** These methods jointly learn a predictive model and an associated interpretation model. This type of work shares similarity with our idea in the sense that they learn interpretable components in an unsupervised manner and leverage them for a transparent reasoning process. However, the interpretable components are learned via an auxiliary task such as image reconstruction instead of directly explored in the learned features (Parekh et al., 2021; Sarkar et al., 2022), making the training process more complicated. Moreover, these models empirically perform worse than only training its predictive black box models by a large margin, which ours guarantees the performance on par with black boxes.

Although we share some similarities with related works, our goal and overall methodology have no overlap with them. Furthermore, to the best of our knowledge, this is the first work constructing a prototype based model in a post-hoc manner, exploring a reverse direction compared to prior works.

## 3 METHOD

In this section, we first introduce our simple 2-step strategy to turn a black box model into a prototype based model with transparent reasoning process. Then we compare the key difference with prior prototype based models. Note that we do not claim to outperform all prior prototype based models in all aspects. Instead, we show in this work that a simple strategy can readily turn a black box model into a prototype based model bringing similar level of interpretability benefits as prior works (e.g., transparent reasoning process based on comparison with interpretable prototypes) while guaranteeing the recognition performance.

### 3.1 STEP1: PROTOTYPE CONSTRUCTION VIA NON-NEGATIVE MATRIX FACTORIZATION

Given $n$ images from class $c$, the $i^{th}$ image's features extracted by neural networks are flattened as $F_i^c \in \mathbb{R}^{HW \times D}$, where $H, W, D$ are height, width and channel dimension number of the extracted feature maps, respectively. The features from $n$ images stacked together are denoted as $A^c = [F_1^c, ..., F_n^c] \in \mathbb{R}^{nHW \times D}$. Matrix factorization tries to decompose $A^c$ into an encoding matrix $E^c \in \mathbb{R}^{nHW \times p}$ and a basis matrix $B^c \in \mathbb{R}^{p \times D}$, where $p$ denotes the number of basis. Denote the

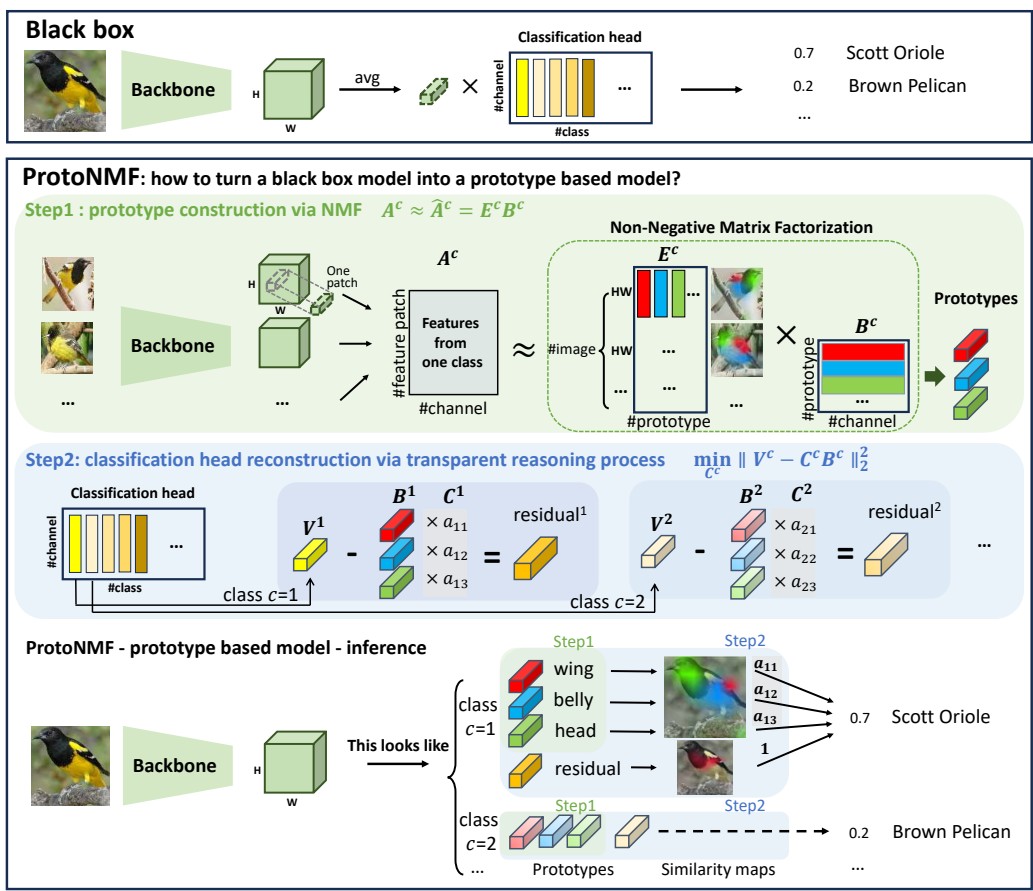

Figure 1: Simple 2-step strategy to turn a black box into a prototype based model. Step 1: calculate prototypes via NMF in features extracted by the black box backbone in each class. Step 2: reconstruct the black box's classification head via discovered prototypes. The importance $a_{11}, a_{12}, a_{13}$ of each prototype is obtained via convex optimization in step 2 and the residual prototype is calculated via the reconstruction difference.

approximate decomposition results as $\hat{A}^c$, the decomposition is expressed as follows:

$$A^c \approx \hat{A}^c = E^c B^c, A^c \in \mathbb{R}^{nHW \times D}, E^c \in \mathbb{R}^{nHW \times p}, B^c \in \mathbb{R}^{p \times D} \qquad (1)$$

In this way, each row of the feature matrix $A^c$ is approximately described as the sum of row vectors in $B^c$ weighted by the values in corresponding rows of the encoding matrix $E^c$, as demonstrated in Figure 1, where different colors in $E^c$ indicate the weight of different row vectors in $B^c$. $E^c$ can also be understood as coordinates of $\hat{A}^c$ in the feature space spanned by the basis vectors (rows) in $B^c$. If we can find a factorization such that the row vectors in $B^c$ are prototypes with parts based representations, we can successfully describe any feature belonging to a certain class as a weighted combination of these prototypes. The columns of $E^c$ with the shape $nHW$ can also be reshaped to $n$ heatmaps with the shape $H \times W$ for visualization, showing how strong and where a certain prototype is present in $n$ images.

To achieve parts based representations, a natural solution is conducting the NMF, which only allows additive, and not subtractive combinations of components, removing complex cancellations of positive and negative values in $E^c B^c$. Such a constraint on the sign of the decomposed matrices is proved to lead to sparsity (Lee & Seung, 2000) and thus lead to parts based representations (Wang & Zhang, 2012), which are therefore suitable to serve as prototypes to indicate individual components of the target object under the "this looks like that" framework. Moreover, an example of interpretability benefit from the non-negative sign is that it's more meaningful to describe a bird as a combination of a head plus the body and a wing instead of a head plus the body minus a wing.

**Efficient multiplicative update** For the ease of reading, we omit the upper index $c$ of $E^c, A^c, B^c$ in this paragraph. NMF is a very efficient parameter-free method for decomposing multivariate data

into strictly positive encoding matrix $E$ and basis matrix $B$. Compared to standard gradient descent, the efficient multiplicative update rules of NMF can guarantee the convergence and no step size parameter is needed. These update rules are computationally inexpensive (Eggert & Korner, 2004). Given a data matrix $A$ with positive values, we randomly initialize the matrices $E$ and $B$ using standard Gaussian distributions while setting all negative values to zero, and then apply the update rules towards optimizing the Euclidean distance $||A - EB||_2^2$ as follows (Lee & Seung, 2000):

$$E_{ij} \leftarrow E_{ij} \frac{(AB^T)_{ij}}{(EBB^T)_{ij}}, B_{jk} \leftarrow B_{jk} \frac{(E^T A)_{jk}}{(E^T EB)_{jk}} \tag{2}$$

The $i, j$, and $j, k$ are indices of corresponding rows and columns in the matrices $E, B$ respectively. The above update rules guarantee that the Euclidean distance $||A - EB||_2^2$ is non-increasing and we refer to (Lee & Seung, 2000) for the convergence proof. The update will stop when the relative error change is small enough (e.g., $10^{-4}$) or maximum iteration times (e.g., 200) is reached. The relative error change is calculated as the error difference between 2 updates divided by the initial error calculated using randomly initialized prototypes, where the error is calculated via:

$$err = ||A - EB||_2^2 \tag{3}$$

**How to handle negative values** CNN architectures (He et al., 2016; Simonyan & Zisserman, 2014) typically use ReLU (Nair & Hinton, 2010) as the activation functions and naturally obtain non-negative features. However, more recent architectures such as ViT (Dosovitskiy et al., 2020) or CoC (Ma et al., 2023) use GeLU (Hendrycks & Gimpel, 2016) as activation functions and thus allow negative feature values, making NMF not directly appliable. We note that there are variants of NMF such as semi-NMF and convex-NMF which could handle negative input values (Ding et al., 2008). However, for consistency in the evaluation across different architectures, we simply set all negative features to zero and conduct NMF on extracted deep features. We demonstrate in later sections that this simple strategy empirically also leads to meaningful parts based representations, probably because most information are already expressed by the non-negative part of the features.

### 3.2 STEP2: CLASSIFICATION HEAD RECONSTRUCTION VIA TRANSPARENT REASONING

A transparent reasoning process requires the final decision to be clearly attributed to the contribution of each interpretable component additively, and the number of components should be small to keep the explanation easily digestable (Alvarez Melis & Jaakkola, 2018). Therefore, we adopt the linear combination of a small number of prototypes for the classification head reconstruction. As shown in Figure 1, we compare the features with each prototype belonging to a certain class and average the weighted sum of similarity scores for the prediction, obtaining a transparent reasoning process. Denote the classification head for class $c$ as $V^c \in \mathbb{R}^{1 \times D}$, we seek for a reconstruction of $V^c$ via a linear combination of $p$ interpretable basis in $B \in \mathbb{R}^{p \times D}$:

$$\min_{C^c} ||V^c - C^c B^c||_2^2 \tag{4}$$

where $C^c \in \mathbb{R}^{1 \times p}$ is the coefficient indicating the importance of prototypes with respect to the class $c$. Since $V^c$ and $B^c$ are fixed, this is a convex optimization and thus $C^c$ has a global optimum. As we constrain the number of prototypes $p$ to be a small number (e.g., p=3) for the ease of human understanding, while the feature dimension is typically high in modern neural networks (e.g., 512 in ResNet34), there is no guarantee that the basis consisting of these $p$ prototypes can precisely reconstruct the black box's classification head. Thus we further introduce the residual prototype as:

$$R^c = V^c - C_{opt}^c B^c \tag{5}$$

where $C_{opt}$ is the optimal coefficient obtained in the optimization objective 4. First optimizing $C_{opt}^c$ and then calculating $R^c$ instead of directly calculating $V^c - B^c$ enables the discovered residual prototype to only represent what the basis $B^c$ can not represent. The interpretability of this residual prototype is visualized via the optimal $H^c \in \mathbb{R}^{nHW}$ through the following convex optimization:

$$\min_{H} ||A^c - E_{opt}^c B^c - H^c R^c||_2^2 \tag{6}$$

where $H^c$ is optimized to optimally reconstruct the features that are not modeled by NMF prototypes ($A^c - E_{opt}^c B^c$) using the discovered residual prototype $R^c$. The obtained $H^c \in \mathbb{R}^{nHW}$ can be

visualized via $n$ heatmaps of the shape $H \times W$, showing where and how strong the residual prototype is present in the features not modeled by NMF prototypes. Surprisingly, these discovered prototypes are not random signals. Instead, as shown in next section, they are mostly present in the foreground, offering insights such as there are still undiscovered discriminative patterns in these areas. Final visualizations are created by upsampling these heatmaps back to the original image resolution for the ease of reading.

### 3.3 OVERVIEW OF THE INFERENCE PROCESS

The interpretability benefit of our method compared to a black box comes from the following classification head decomposition during the inference (an illustrative figure is offered in A.3):

$$V^c = C_{opt}^c B^c + R^c = a_1 b_1 + a_2 b_2 + ... + a_p b_p + R^c \qquad (7)$$

where $a_i$ are the scalers in $C_{opt}^c \in \mathbb{R}^{1 \times p}$ indicating the importance of each prototype, $b_i \in \mathbb{R}^{1 \times D}$ indicates the $i^{th}$ NMF prototype obtained in $B^c \in \mathbb{R}^{p \times D}$. Since the residual prototype might be less intperpretable, we further propose the following strategy to distribute the residual prototype's parameters to $p$ NMF prototypes and remove the explicit inference dependence on it:

$$V^c = a_1 (b_1 + \frac{R^c}{\sum_{i=1}^p a_i}) + ... + a_p (b_p + \frac{R^c}{\sum_{i=1}^p a_i}) \qquad (8)$$

Similar to equation 1, the visualization of $p$ augmented prototypes in the form of $b_i + \frac{R^c}{\sum_{i=1}^p a_i}$ could be obtained via $E^{c'} \in \mathbb{R}^{nHW \times p}$ in the following convex optimization:

$$\min_{E'} ||A^c - E^{c'} B^{c'}||_2^2 \qquad (9)$$

where the rows of $B^{c'} \in \mathbb{R}^{p \times D}$ are the augmented NMF prototypes $b_i + \frac{R^c}{\sum_{i=1}^p a_i}$. We refer to the section A.2 of appendix to show that the augmented prototypes are still interpretable.

### 3.4 COMPARION WITH A PREVIOUS PROTOTYPE BASED MODEL

The key difference of our ProtoNMF compared to prior prototype based methods lives in the prototype obtaining process. We take ProtopNet (Chen et al., 2019) as our major baseline as other following works employ the same prototype obtaining process. ProtopNet (Chen et al., 2019) includes a periodic three-stage training scheme: (1) fix the final classification head and train the prototype layer as well as the backbone. (2) replace the latent vectors in prototype layer with closest feature patch from the training set. (3) fine-tuning the final classification head. Three stages are repeated multiple times until convergence.

It's straightforward to see that in the second stage, each prototype corresponds to the feature of a single image patch. However, this may make the meaning of the prototype ambiguous (e.g., is the shape or the texture or the color of this patch important?). In contrast, the meaning of our prototype can be identified via observing the shared features among a set of images, making it easier for human to understand. In addition, we note that our latent representations of prototypes may have other advantages, such as being more stable and comprehensive during the training compared to ProtopNet (Chen et al., 2019). It's also easy to change the prototype number to offer richer explanations, while prior models must be retrained to achieve this.

## 4 EXPERIMENTS

The experiments are conducted in both fine-grained (e.g., CUB-200-2011 (Wah et al., 2011)) and general classification datasets (e.g., ImageNet (Deng et al., 2009)), where CUB-200-2011 (200 bird spieces) (Wah et al., 2011) is a benchmark dataset in prototype based models and ImageNet (1000 classes) (Deng et al., 2009) is less evaluated by prior prototype based models. In addition, we demonstrate qualitatively that our proposed ProtoNMF also yield interpretable prototypes in more recent architectures such as ViT (Dosovitskiy et al., 2020) and CoC (Ma et al., 2023). Extensive evaluations and in-depth analysis are offered discussing the interpretability and the performance. We follow (Chen et al., 2019) in CUB preprocessing/augmentation and leverage publicly available checkpoints in the ImageNet (Deng et al., 2009) for analysis.

Table 1: NMF obtained prototypes serve as better basis towards a lower class features' reconstruction error. The values are the mean and standard deviation of the error across all classes and 3 runs.

| Training cycle | ProtopNet | NMF |
|---|---|---|
| Cycle 1 | 43.5±10.2 | **24.9**±6.5 |
| Cycle 2 | 99.0 ±10.6 | **25.7**±1.4 |
| Cycle 3 | 110.2±14.8 | **25.1**±2.7 |

Table 2: Performance comparison in CUB-200-2011 (Wah et al., 2011).

| Methods | Acc |
|---|---|
| ProtoPNet (Chen et al., 2019) | 79.2 |
| ProtopShare (Rymarczyk et al., 2021) | 74.7 |
| ProtoPool (Rymarczyk et al., 2022) | 80.3 |
| Def. ProtoPNet (Donnelly et al., 2022) | 76.8 |
| ProtoKNN (Ukai et al., 2022) | 77.6 |
| ResNet34 (He et al., 2016) (ProtoNMF) | **82.3** |

## 4.1 CUB-200-2011

**How comprehensive is the reasoning base?** Since it's even hard to compare whether one set of prototypical images selected by one expert is more comprehensive than another expert considering the unlimited variations in angles, lighting conditions and bird actions, we compare whether a set of prototypes is more comprehensive than another set quantitatively in the latent space. Assume all features from all images of a class build a complete description of that class, we measure how well a set of prototypes could serve as the basis to reconstruct all features belonging to a certain class via the feature reconstruction error. Concretely, we leverage the equation 3 and compare the optimal reconstruction errors using NMF's calculated prototypes and ProtopNet's Chen et al. (2019) selected prototypes. We use 10 prototypes in both settings for fair comparisons (default setting of ProtopNet). The Table 1 demonstrates quantitatively that prototypes obtained via NMF consistently leads to a lower reconstruction error in different training cycles (one cycle is one iteration of 3 training stages) of the ProtopNet. This indicates that they serve as better basis vectors to reconstruct all features belonging to a class. We evaluate the first 3 cycles because we empirically observe that the checkpoint with the best validation accuracy could possibly appear in any one of the first 3 cycles. Figure 2 demonstrates qualitatively that the prototypes obtained by our ProtoNMF are more comprehensive (e.g., diverse and complete). For the ease of visualization, we take 3 prototypes as an example: in the last two rows, the areas colored green, blue and red are easily separable and consistently indicate the bird head, belly and wing in all images. Besides, the areas corresponding to these three prototypes cover the bird's whole body in a more complete manner compared to the areas indicated by the yellow bounding boxes in the first two rows. Over the course of training, ProtopNet's prototypes are becoming less diverse while ProtoNMF's prototypes remain diverse.

**How stable is the reasoning base?** For fair comparison, we generate ProtoNMF visualizations based on the ResNet34 checkpoints under the same epochs of ProtopNet (Chen et al., 2019). These checkpoints are trained using the same learning rate schedule as ProtopNet. We demonstrate the stability qualitatively in Figure 2: compare the first two rows with the latter two rows, it is evident that ProtopNet exhibits significant variations in the selected prototypes throughout different cycles. Conversely, our ProtoNMF showcases only slight differences in the prototypes' present areas.

**Performance** For fair comparison, all performances are reported based on the backbone ResNet34, as it is commonly evaluated in prior papers. Table 2 shows that ResNet34 outperforms all prior prototype based models. Since our ProtoNMF is a precise reconstruction of ResNet34, ProtoNMF enjoys the same performance.

## 4.2 IMAGENET

Prior works may have different reasons not to evaluate in this large scale dataset with very distinct classes. For example, the idea of sharing prototypes may be less meaningful when classes are very different (Rymarczyk et al., 2021) and Ukai et al. (2022) points out that the computation load may be too heavy for their model. Since comprehensive evaluation of all prior works is out of the scope of this paper, we focus on the in-depth analysis of our proposed method in this dataset and compare with our major baseline ProtopNet (Chen et al., 2019), which do not suffer from above difficulties. We evaluate multiple architectures including ResNet34 (He et al., 2016), ViT (ViT-base with patch size 32) (Dosovitskiy et al., 2020) and CoC (Coc-tiny) (Ma et al., 2023). A brief summary of different architectures is shown in the Figure 3. We fix $p = 3$ in the comparison for ease of understanding unless otherwise indicated.

**Qualitative comparison of NMF prototypes in 3 architectures** The areas that our ProtoNMF are present are visualized in Figure 3 by different colors. It could be seen that even if we remove the negative feature values in ViT and CoC, NMF still leads to reasonable parts based representation.

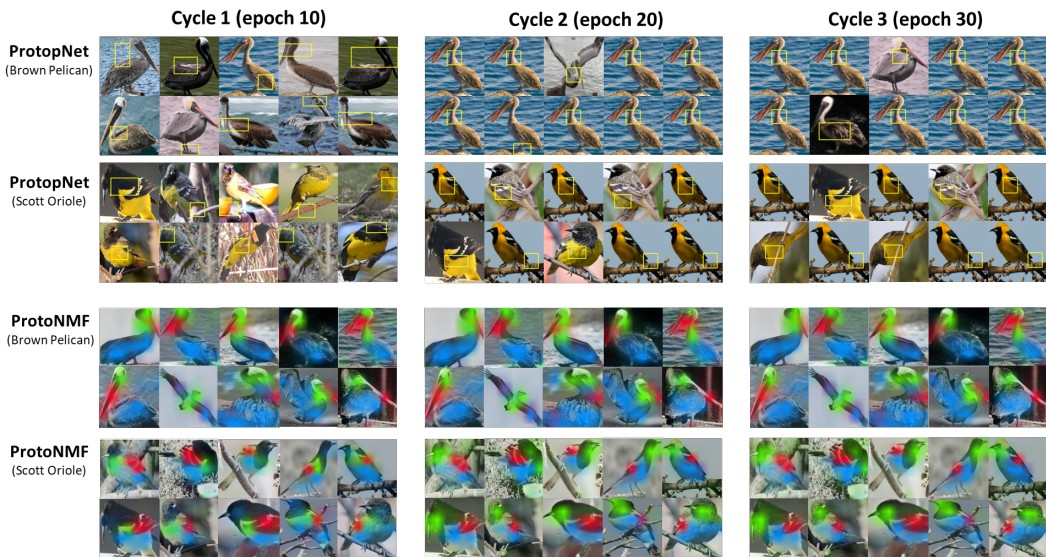

Figure 2: In the first two rows, 10 bounding boxes indicate 10 prototypes selected by ProtopNet (Chen et al., 2019) in each cycle for one class. In the last two rows, each color indicates a different discovered prototype of our ProtoNMF. We randomly choose 10 images to indicate the consistency and diversity of prototypes across images. During the training, the selected prototypes of ProtopNet has a large difference across cycles and tend to indicate the same semantic meaning in later cycles.

This indicates that the non-negative feature values already capture the major representations of the corresponding image parts. Moreover, the areas of different prototypes from ViT are less separable, probably because the ViT architecture has less inductive bias. This may make the spatial relationship in the original image less maintained in the feature map.

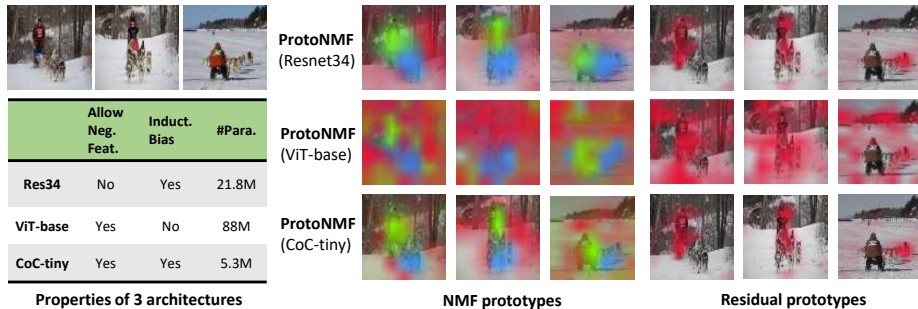

Figure 3: Comparison of 3 architectures and their prototypes in ImageNet (Deng et al., 2009). We visualize the presence areas of 3 prototypes in 3 example images from the class "sled dog".

**Performance comparison** We run the ProtopNet (Chen et al., 2019) using the officially released code in ImageNet (Deng et al., 2009). The Protopnet is trained using an ImageNet pretrained ResNet34 (He et al., 2016) as the backbone. ProtopNet exhibits a strong performance drop in terms of top1 accuracy (65.5) while our ProtoNMF can maintain the performance of ResNet34 (75.1).

**What's the influence of different number of NMF prototypes?** As could be seen in the Figure 5, more NMF prototypes would lead to both smaller feature reconstruction error and classification head reconstruction error in all architectures. However, an limitation is that a larger number may also make the visualization harder for human to interpret, as human generally won't segment an image into too many parts for classification, as shown in Figure 4 (Figures generated using CoC-tiny).

**How discriminative are discovered prototypes?** Table 3 shows the performance of only using NMF/residual/augmented NMF prototypes for classification. Although the presence areas of NMF

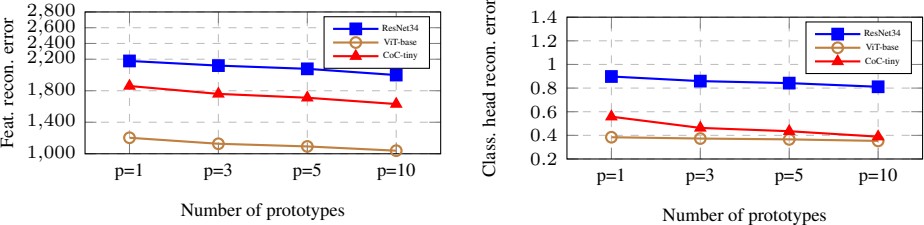

Figure 4: Visualization using different number of prototypes (e.g., p=1,3,5,10.). In each image, a different color indicates the presence area of a different prototype.

Figure 5: More NMF prototypes reduce both feature and classification head reconstruction errors.

prototypes enjoy semantic meanings due to parts based representations, using them alone for the classification is not discriminative enough. This may due to the fact that the pretrained feature extractor is optimized towards the pretrained classifier allowing negative values, and generally a basis consisting of small number of non-negative prototypes may not be precise enough to recover the discriminative capability of the original high-dimensional classification head. Thanks to the parameter distribution strategy, the augmented NMF prototypes are both discriminative and interpretable.

Table 3: Performance using only NMF/residual/augmented NMF prototypes. Note that the result 76.3 of ViT comes from averaged features of the last layer into the classification head for fair comparison. Using the ViT's classification token reaches 80.7 Top1 accuracy.

| **ResNet/ViT/CoC** | Origin. NMF only | Residual only | NMF+residual/Aug. NMF |
|---|---|---|---|
| p=1 | 1.3 / 55.1 / 50.7 | 74.8 / 80.1 / 71.7 | 75.1 / 76.3 / 71.9 |
| p=3 | 16.4 / 50.8 / 45.1 | 72.6 / 79.1 / 60.3 | 75.1 / 76.3 / 71.9 |
| p=5 | 20.4 / 52.1 / 49.3 | 72.0 / 78.5 / 56.9 | 75.1 / 76.3 / 71.9 |
| p=10 | 28.4 / 54.8 / 54.3 | 71.0 / 77.3 / 48.3 | 75.1 / 76.3 / 71.9 |

**How to interpret coefficients of $C_{opt}^c$ during classification head reconstruction?** As a case study, we examine the class "sled dog" in Figure 3. The raw coefficients we obtained for the red, green and blue prototypes (corresponding to background, human, dog) are $-0.004, -0.007, 0.014$ for ResNet34, $0.0007, -0.00006, 0.01$ for ViT-base and $-0.0176, 0.016, 0.019$ for CoC-tiny. Interestingly, these numbers indicate that in ResNet34 and ViT-base architectures, the dog prototype is the most discriminative one, although three interpretable prototypes are found. These numbers also reveal that CoC-tiny's reasoning strongly relies on both human and dog prototypes (0.016 vs 0.019), which may be a correlation bias as sled dogs are often captured sledding for human. In addition, it also relies on the negative reasoning process of background, saying an image is more likely to be a sleg dog if it looks less similar to the background. In general, our method could find which prototypes are more discriminative among all interpretable prototypes, offering a more comprehensive understanding of the reasoning process. This may potentially ease the human debugging/debiasing or test-time intervention of the model. We refer to the appendix A.4 for an example usage.

## 5 CONCLUSION

Prior prototype based interpretable models typically under-perform black box models. Therefore, we take the first step to explore the reverse direction and aim to answer: can we turn a trained black box model into a prototype based model while discovering interpretable prototypes for transparent reasoning? To this end, we propose a simple 2-step strategy to first construct prototypes from backbone's extracted features via NMF and then reconstruct the classification head via discovered prototypes. We demonstrate competitive interpretability benefits of this approach (e.g., offer transparent reasoning with meaningful prototypes) while guaranteeing the recognition performance. Extensive evaluations are offered to discuss both benefits and limitations of the method.

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

## A APPENDIX

### A.1 IMPLEMENTATION DETAILS

The implementation details of step 1: prototype reconstruction via NMF are already described in the method part of the main text and we refer to Collins et al. (2018) for the NMF code base. Here we offer the code for step 2: classification head reconstruction via transparent reasoning. The obtained W or residual-coeff may be bilinearly interpolated back to the resolution of the original input images for visualization of NMF prototypes and residual prototypes.

```python
import torch
from nmf import NMF
from utils import imresize, show_heatmaps
import cvxpy as cp
def decomposition(cls_idx, flat_features,features, model,raw_images, K):
    with torch.no_grad():
        #Calculate NMF prototype
        W, H, feat_rec_error = NMF(flat_features, K, random_seed=0, cuda=False, max_iter=200,verbose=True)
        W_coeff = cp.Variable(K)
        objective = cp.Minimize(cp.sum_squares(W_coeff@H -model.get_classifier().weight[cls_idx]))
        prob = cp.Problem(objective)
        result = prob.solve()
        print("class",cls_idx, "feature recon error:",feat_rec_error, "class vector recon error:", result,  "K=",K)
        # H indicates the constructed prototypes via NMF, W indicates how strong and where they are present.
        # W_coeff indicates the importance of each prototype regarding each class (C_{opt}^c in the main paper)

        #Interpretation of the residual prototype
        feature_diff = flat_features - torch.mm(W, H)
        residual_diff = model.get_classifier().weight[cls_idx] - torch.matmul(torch.from_numpy(W_coeff.value).float(),H)
        residual_diff = residual_diff.unsqueeze(0)
        residual_coeff = cp.Variable((flat_features.shape[0],1))
        objective = cp.Minimize(cp.sum_squares(residual_coeff@residual_diff - feature_diff))
        prob = cp.Problem(objective)
        result = prob.solve()
        #residual_coeff indicates where the residual prototype is present in the features not modeled by NMF prototypes.
    return H, W_coeff
```

Figure 6: Code for classification head reconstruction.

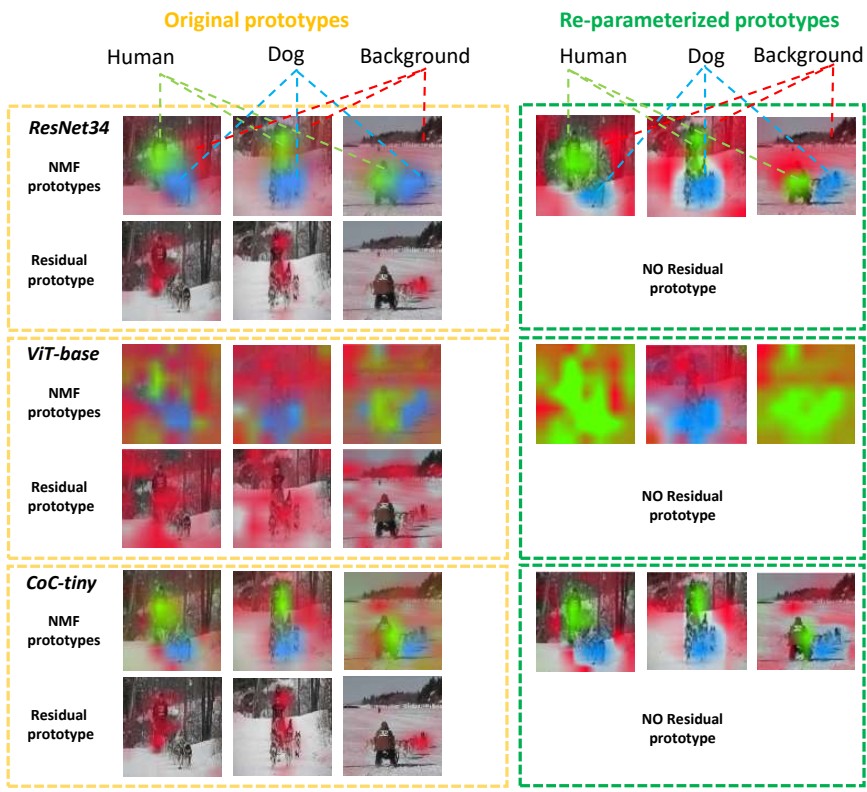

Figure 7: Interpretability comparison between original NMF prototypes and re-parameterized NMF prototypes.

### A.2 INFLUENCE OF RE-PARAMETERIZATION ON THE INTERPRETABILITY

**Empirical evidence** Fig. 7 shows that the prototypes with enhanced discriminative power still deliver interpretable parts based representations. In architectures with inductive bias such as ResNet34 and CoC-tiny, the re-parameterized prototypes in the foreground become even better separated from the background prototype, compare blue/green areas with red areas.

**Theoretical analysis** We focus the analysis on the difference between encoding matrix $E^c$ and $E^{c'}$ for interpretability visualization. Observe the equation 8 for re-parameterization, we are actually shifting every axis of the coordinate system consisting of $p$ prototypes by the vector $\frac{R^c}{\sum_{i=1}^{p} a_i}$. Therefore, the change in the interpretability visualization is reflected by the change of the coordinates of all features in this coordinate system. For the ease of analysis, first assume each prototype is orthogonal to each other. Consider any prototype $b_i$, the strength of the $m^{th}$ feature patch $A_m^c$ from class c along the $k^{th}$ prototype $b_k$ is $A_m^c b_k^T$ before re-parameterization. After the re-parameterization, the projected strength becomes $A_m^c b_k^T + A_m^c (\frac{R^c}{\sum_{i=1}^{p} a_i})^T$. Therefore, the change of the visualization completely depends on the term $A_m^c (\frac{R^c}{\sum_{i=1}^{p} a_i})^T$. Fortunately, it's reasonable to expect this term to be very small compared to the first term $A_m^c b_k^T$ especially in the salient areas where the first term is large and contribute the most to the interpretability visualization. This is because original NMF prototypes $B^c$ are explicitly optimized to approximate the features $A^c$. Though we use the assumption that prototypes are strictly orthogonal to derive this conclusion, a similar logic also applies to not strictly orthogonal NMF prototypes, because NMF is exactly an optimization process to make the prototypes as diverse/orthogonal as possible, yet serving as a good basis to span a feature space covering the features of the target class.

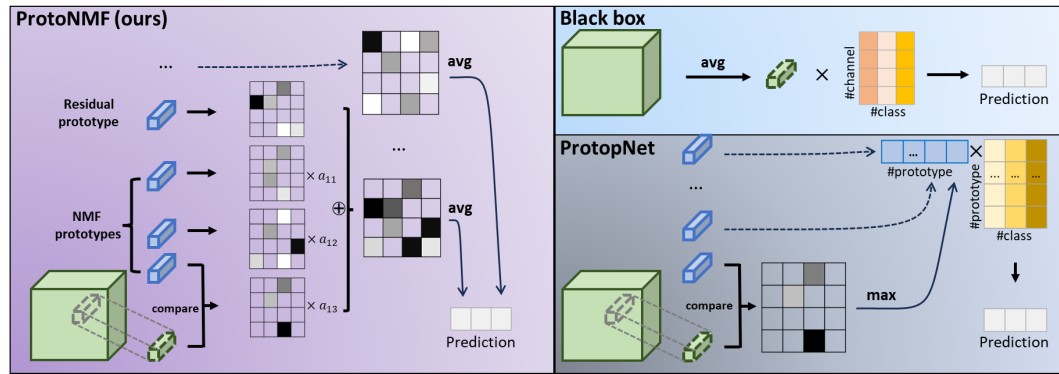

Figure 8: Comparison of the reasoning process of our ProtoNMF, ProtopNet and black box.

## A.3 COMPARISON OF THE REASONING PROCESS

Both our ProtoNMF and the baseline ProtopNet follows the reasoning process based on the comparison with a set of prototypical parts, while the black box compares the features with a single class vector for the classification logit of each class. A slight difference between the ProtoNMF and ProtopNet is that after obtaining the similarity map of each feature patch to a prototype, we average the similarities of the map and linearly combine these similarities of all prototypes belonging to a class for the class logit, while the ProtopNet takes the maximum value of the similarity map and linearly combine these similarities. Another difference is that ProtopNet leverages the prototypes of all classes for the class logit of a single class, while we only leverage a small set of prototypes belonging to each individual class. However, we argue that these slight differences (e.g., average versus maximum, less versus more prototypes) do not influence the fact that the overall inference framework is still based on the comparison with prototypes.

## A.4 CASE STUDY OF PROTOTYPE BASED TEST-TIME INTERVENTION

The figure 9 demonstrates an example of how experts could leverage the prototypes to conduct test-time intervention to correct the model's prediction. (1) The red prototype corresponds more to the background. (2) The green prototype is more in the shape of a long thin pipe, which confuses the model when the input is an instrument with the similar shape. (3) The blue prototype corresponds more to a trigger. So a human expert could intervene the green prototype by setting the contribution from this prototype to zero and the model can now correctly predict the input image to obe, hautboy, hautbois.

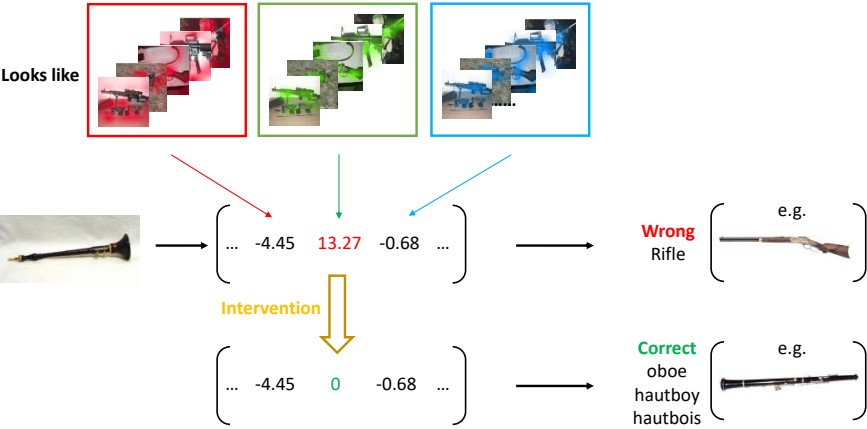

Figure 9: A successful example of test-time intervention.

**Quantitative evaluation** As explained in section 1, $5^{th}$ paragraph of ProtopNet Chen et al. (2019), even if some datasets have interpretable attribute annotations, since the discovered concepts by prototype based models may not exist in the existing dataset's pre-defined attributes, it's hard to evaluate quantitatively how good are the discovered prototypes. However, we try to offer some quantitative results to bring more insights of the advantage of our method compared to a black box under some assumptions. Assume in all wrong predictions, an expert exist that is allowed to intervene one prototype for one time in each prediction, and this expert always intervenes the prototype with highest importance of the predicted class (indicated by $\max(C_{opt}^c)$) by setting the contribution of this prototype to 0, the Top1 accuracy of the model will increase by $10.8\%$ (Tab. 4). Note that in practice,

Table 4: Performance of CoC-tiny before and after human intervention in ImageNet.

| Top1 Acc without intervention | Top1 Acc with intervention |
|---|---|
| 71.9 | 82.7 |

experts may apply a much more complicated intervention strategy such as multiple interventions or intervene multiple prototypes simultaneously to further increase the performance. We leave designing effective and efficient intervention strategy to future work.

