# OpenReview forum: "ProtoNMF: Turning a Black Box into a Prototype Based Interpretable Model via Non-negative Matrix Factorization"
_ICLR.cc/2024/Conference — Submitted to ICLR 2024_

### Official Review · Reviewer_XoPR · 2023-10-28

**Soundness:** 3 good
**Presentation:** 2 fair
**Contribution:** 3 good
**Rating:** 5
**Confidence:** 3

**Summary:**

The paper proposes a post-training decomposition technique for learning prototypes for interpretable image classification. Specifically, the paper leverages Non-negative Matrix Factorization (NMF) to learn prototypes (bases) for a certain class from a batch of hidden image features from the same class. The prototypes are then used to reconstruct the learned feature classifier for this class. By visualizing the attention on the prototypes, researchers can identify interpretable regions and their importance in arriving at the final classification results. The paper demonstrated good interpretability in the experiment.

**Strengths:**

* **Simple post-training solution**: the paper proposed a simple solution to enable better interpretability for a trained model without modifying the training process. Compared to prior works, the proposed method is computationally efficient and is architecture-agnostic.

* **Good interpretability and classification accuracy**: while prior works often sacrifice classification accuracy because of the modification to the training pipeline, the proposed model brings interprtablity without loosing accuracy.

* **Detailed analysis**: the paper provides a detailed empirical analysis of various aspects of the model, including the discriminativeness of the extracted prototypes, which is also a limitation of the method.

**Weaknesses:**

* **Lacking an inference description**: the paper lacks a discussion on the inference procedure in the method section. While Figure 1 provides schematics, it is not clear enough. My understanding is the following: the paper uses the *original* head classifier for classification because
$$V^c = R^c + C^c_{opt}B^c$$
where $V^c$ is the original classifier vector, $R^c$ is the residual prototype and $ C^c_{opt}B^c$ is the extracted prototypes. The paper uses both the residual and the extracted prototypes, the sum of which amounts to the original classifier. This is equivalent to using the original classifiers for classification. This is the reason why the proposed method guarantees no drop in accuracy.

* **Extracted prototypes not discriminative**: the paper provides a detailed analysis of the discriminativeness of the extracted prototypes $ C^c_{opt}B^c$. The conclusion is that they are not discriminative enough (if at all when the number of prototypes is small according to Table 3) .This makes one wonder if this discovery defeats the main purpose of the paper: discovering meaningful prototypes and shedding light on a transparent reasoning process, because these prototypes are neither meaningful nor explaining the model's decision for a specific class. The fact that using the extracted prototypes alone results in poor classification accuracy makes one think that the proposed NMS procedure is ineffective in extracting good prototypes for classification.

**Questions:**

* Can the authors comment on my concerns regarding the meaningfulness and usefulness of the extracted prototypes in the weakness section?

* A follow-up question is how important the discriminativeness of the prototypes is in interpreting the decision-making process in classification and what information we would miss if the prototypes were not discriminative as in the proposed method.

I really like the proposed method but the main concern regarding the discriminativeness of the prototypes also weighs heavily in my decision. I will be happy to raise my score if the authors can address it convincingly.

---

> ### Author Response · Authors · 2023-11-21
> **Response to reviewer XoPR**
>
> Thank you so much for liking our paper and we are highly encouraged by your positive and constructive feedback! It is our pleasure to address the primary concerns in the following sections in addition to the general rebuttal given above.
>
> **Adding an inference description** Thank you for pointing this out. Your understanding is completely correct and we fully agree that it is helpful for readers to better understand our method. We have revised the section 3 of the manuscript to include this part.
>
> **How important the discriminativeness of prototypes is** We agree that if the extracted prototypes are less discriminative, one may challenge their roles in the decision making. Thus we further propose a simple re-parameterization (detailed in general response above) to remove the residual prototype and increase the discriminative power of NMF prototypes, yet maintaining the interpretability of these prototypes.
>
> **What is missing in original NMF prototypes?** A straightforward answer is the parameters in the residual prototype, because empirical evidence shows adding them can reach the original performance. A more intuitive understanding is: the features extracted by the pre-trained feature extractor are optimized to fit the pre-trained classifier, thus a precise reconstruction of the pre-trained classifier becomes important in maintaining the discriminativeness. However, in the general case, it's hard to obtain a precise reconstruction of a high-dimensional (e.g., 320 for coc-tiny, 512 for res34, 768 for vit-base) classification head using a basis consisting of small number of prototypes (e.g., 1,3,5,10), which is why the approximated head using only NMF prototypes does not perform well. Therefore, it's important that the missing part after the optimal reconstruction could be incorporated into the existing basis if one wants to express the classification head using exactly $p$ interpretable prototypes. The above analysis largely inspired the re-parameterization idea and we are very grateful to your in-depth and insightful comments!
>
> We hope our rebuttal could address your concerns and sincerely hope that the reviewer could consider increasing the score.

---

### Official Review · Reviewer_A726 · 2023-10-31

**Soundness:** 2 fair
**Presentation:** 2 fair
**Contribution:** 2 fair
**Rating:** 5
**Confidence:** 3

**Summary:**

In this paper, the authors proposed ProtoNMF, a method to turn a black-box model into a prototype-based interpretable model using non-negative matrix factorization (NMF). The method involves constructing a feature matrix A^c for a given class c by stacking the D-dimensional feature vectors of n*H*W image features from n images as rows, and applying NMF to A^c to yield the factorization A^c = E^c B^c where B^c is the prototype matrix whose rows are D-dimensional prototype basis vectors, and E^c is the encoding matrix whose rows represent the coordinates of the image features along the prototype basis vectors. The method also involves another step to reconstruct a classification head V^c for a given class c, using a linear combination of prototype vectors (rows) of B^c, and to find a residual prototype R^c = V^c - C^c_opt B^c, where C^c_opt B^c is the best linear combination of prototype vectors (rows) of B^c that approximates the classification head V^c. The computation of the logit of class c of the original black-box model on the image features A^c can then be thought of as first computing a linear combination of prototype vectors in B^c (i.e., E^c_opt B^c), and then adding scalar multiples of the residual prototype R^c to each spatial position of each image (i.e., H^c_opt R^c). The authors conducted experiments on CUB-200-2011 and ImageNet to demonstrate the efficacy of their ProtoNMF.

**Strengths:**

- The proposed ProtoNMF can preserve the accuracy of a black-box model.

**Weaknesses:**

- The proposed ProtoNMF cannot be interpreted in the same way as the baseline ProtoPNet. Its interpretability is far from ProtoPNet. The prototypes are not constrained to be actual image features of some training images. How are they visualized?
- The proposed ProtoNMF uses linear combinations of prototypes, rather than similarities to prototypes. This, again, reduces interpretability of ProtoNMF. What do linear combinations of (abstract) prototype vectors even mean?
- The proposed ProtoNMF also relies on a residual prototype for each class. Again, the interpretation of a "residual prototype" is unclear.

**Questions:**

- As mentioned earlier, the prototypes from ProtoNMF are obtained via NMF and are not constrained to be actual image features of some training images. How are the prototypes visualized?
- As mentioned earlier, ProtoNMF uses linear combinations of prototypes. What do linear combinations of (abstract) prototype vectors even mean?

**Details Of Ethics Concerns:**

N/A.

---

> ### Author Response · Authors · 2023-11-20
> **Response to reviewer A726 (Part I)**
>
> We express our deepest gratitude for the reviewer’s time and constructive comments. It is our pleasure to address the primary concerns in the following sections in addition to the general rebuttal given above.
>
> Before addressing the reviewer's concern, we want to clarify two miss-understanding points based on the reviewer's summary.
>
> **Computation of the logit of class c** is not based on the $E^c_{opt} B^c$, but is based on the $C^c_{opt}B^c$ [equation 4,5]. $C^c_{opt}$ are the coefficients optimized to make the NMF prototypes approximate the black box's classification head $V_c$ and the $E^c_{opt} \in \mathbb{R}^{nHW \times p}$ is used to indicate the interpretability of $B^c$ [section 3.1, 1st paragraph, last sentence].
>
> **The residual prototype** is not added into each spatial position of each image via $H^c_{opt} R^c$. The $H^c_{opt} \in \mathbb{R}^{nHW\times 1}$ is used to indicate where and how strong the residual prototype is present in n images [section 3.2, paragraph below equation 6]. As pointed out by reviewer XoPR, the final classification head $V^c\in \mathbb{R}^{1\times D}$ decomposition can be expressed via rewriting the equation 5 as:
> \begin{equation}
>     V^c = C_{opt}^cB^c+R^c
> \end{equation}
>
> We apologize for causing the miss-understanding and have improved our writing in section 3 of the revised manuscript. We also added the Figure 8 to the appendix to compare the inference process of the black box, ProtopNet and our ProtoNMF.
>
> **How are latent prototypes which are not directly the feature of any image patch visualized** It is visualized via indicating how important this prototype is to reconstruct the feature of each position of a set of real images. Given one prototype, such importance scores in all positions of one image together build one visualization. And combining such visualizations of all images makes human even easier to understand what exactly is important in these areas by observing common features in salient areas across all images.
>
> **Visualization of NMF prototypes** The $E^c_{opt} \in \mathbb{R}^{nHW \times p}$ is used for visualization [section 3.1, $1^{st}$ paragraph, last sentence]. As a simple example, the first row in $E^c_{opt} \in \mathbb{R}^{nHW\times p}$ represents the $p$ coefficients of the first image patch (e.g., the upper left patch of an image) of the first image, and the first coefficient in this row indicates to which degree the feature of this patch can be approximated/reconstructed by the first prototype. So a large value in the $E^c_{opt}[1,1]$ means the first patch's feature is very similar to the first latent prototype. Combining the first $HW$ values of the first column tells us how similar each patch of the first image is to the first prototype. A visualization is shown in the dashed line's area of step 1 of Figure 1 of the manuscript, where the strength of the red color in the first bird image correspond to the first $HW$ red values in the first column of the $E^c$.
>
> **Visualization of the residual prototype**. The visualization is similar. The only difference is: instead of looking at how similar the residual prototype is to the image patch's features, we look at how similar the residual prototype is to the image patch's "residual" feature. The "residual" feature indicates the feature that can not be modeled by the NMF prototypes [section 3.2, paragraph under equation 6]. For both types of prototypes, we apply a bilinear interpolation to upsample the resolution of the feature map to the resolution of the original image [section 3.2, last sentence]. At last but not least, we want to point out that a recently published paper by the ProtopNet's author also leverages the latent prototype and visualize them via a set of images instead of a single image patch [1].
>
> [1] Chiyu Ma, Brandon Zhao, Chaofan Chen, Cynthia Rudin. This Looks Like Those: Illuminating Prototypical Concepts Using Multiple Visualizations, NeurIPS 2023.

---

> > ### Author Response · Authors · 2023-11-20
> > **Response to reviewer A726 (Part II)**
> >
> > **Similarity comparison based inference \& meaning of linear combination of prototypes** We are indeed using the similarity to each individual prototypes in the inference. For simplicity, assume we are given one feature $X\in \mathbb{R}^{1\times D}$ representing the whole image (e.g., the average of the final feature map), the classification logit of class $c$ is calculated as $XV_c^T \in \mathbb{R}$. After decomposing the $V^c$ into $C_{opt}^cB^c$ and $R^c$, the classification logit for class $c$ becomes
> > \begin{equation}
> > XV_c^T = X(C_{opt}^cB^c+R^c)^T = X(C_{opt}^cB^c)^T + X(R^c)^T
> > \end{equation}
> > The vector-matrix multiplication between $C_{opt}^c \in \mathbb{R}^{1\times p}$ and $B^c\in \mathbb{R}^{p \times D}$ can be further decomposed into a linear combination of vectors. Therefore, $C_{opt}^cB^c$ is _not understood as one whole un-interpretable prototype_, but just a matrix expression of a set of individual interpretable prototypes. Thus the final inference is based on the feature $X$'s comparison to each individual interpretable prototypes as follows:
> > \begin{equation}
> > X(C_{opt}^cB^c)^T + X(R^c)^T = a_1Xb_1^T + a2Xb_2^T + ... +a_pXb_p^T + X(R^c)^T
> > \end{equation}
> > where the $a_i$ is the scaler in the vector $C_{opt}^c \in \mathbb{R}^{1\times p}$. These scalers indicate the general importance of each individual prototype $b_i$ for class c. The similarity of the feature $X$ to each prototype is $X b_i^T \in \mathbb{R}$, explaining why a specific input is classified to some specific class. In addition, we note that the baseline ProtopNet [2] and nearly all following works have a linear layer combining prototypes after the prototype comparison layer. The usage of linear combination has the interpretability benefit in that the contribution of each component can be clearly identified. This is also commonly accepted in literature of other interpretable methods not based on prototypes [3,4]. In addition, we offer the Figure 8 in the appendix for a more illustrative explanation of the prototype based inference process. The section 3.3 of the manuscript is also accordingly updated for easier understanding of the inference process.
> >
> > [2] Chen, Chaofan, et al. "This looks like that: deep learning for interpretable image recognition." Advances in neural information processing systems 32 (2019).
> >
> > [3] Alvarez Melis, David, and Tommi Jaakkola. "Towards robust interpretability with self-explaining neural networks." Advances in neural information processing systems 31 (2018).
> >
> > [4] Radenovic, Filip, Abhimanyu Dubey, and Dhruv Mahajan. "Neural basis models for interpretability." Advances in Neural Information Processing Systems 35 (2022): 8414-8426.
> >
> > **Interpretability of the residual prototype**
> > We refer to the common response to all reviewers, where a simple re-parameterization could remove the residual prototype and increase the discriminative power of NMF prototypes, yet maintaining the interpretability of these prototypes.
> >
> > We hope our rebuttal could address your concerns and sincerely hope that the reviewer could consider increasing the score.

---

### Official Review · Reviewer_eeCj · 2023-11-01

**Soundness:** 2 fair
**Presentation:** 2 fair
**Contribution:** 2 fair
**Rating:** 3
**Confidence:** 2

**Summary:**

This paper introduces a method that turns a black-box pretrained image classification network into a more interpretable prototype-based network, by performing non-negative matrix factorization to decompose the final features of the network into non-negative linear combinations of bases of classes. The authors claim that in this way, the model can achieve interpretability without sacrificing performance. Empirical evaluations are performance on two datasets and three difference model architectures.

**Strengths:**

1. The comprehensive evaluations of different model architectures are appreciated.

2. The idea of turning a black box model into a more interpretable one is interesting.

**Weaknesses:**

1. Interpretability

I believe the main contribution of this paper is to improve the interpretability of a pretrained black-box model. However, after reading the paper, I have no idea how to measure the improvements in interpretability quantitatively. The visualization of the prototypes may show how the model makes the final prediction, but I believe regular 'black-box' networks + GradCAM can do the same and there is no obvious evidence of the advantage of the proposed method.
One thing the author mentioned is that such prototypes can help the post-training human intervention in the model. However, the missing of this part in the experiment section makes it very hard to justify the contribution of 'interpretability.'

And I am afraid that the 'residual prototypes,' which seem to be crucial for maintaining the recognition performance, will make it even harder to intervene manually in the model.

In summary, the authors are expected to do more than visualizations to support the interpretability.

2. Writing and presentation

The overall writing of this paper is relatively casual and in many cases not precise enough for the readers to properly learn the ideas.
And some examples are not solid enough.
For example, in Figure 2, it is true that ProtopNet is clearly converging to fewer prototypes as the training goes longer, the visualization of the learned prototypes of ProtoNMF on different images does not show the superiority in terms of diversity. How distinct are those learned prototypes?

3. Evaluations

Please consider adding the performance of the standard ResNet34 to table 2.

And I personally believe the results in Table 3 do not support the claim 'guarantee the performance on par with the black box models.' In most the cases, the performance decreases drastically. And the best performance is with the not-so-interpretable residuals.

**Questions:**

Please evaluate the interpretability of the proposed method quantitatively.

---

> ### Author Response · Authors · 2023-11-20
> **Response to reviewer eeCj**
>
> We express our deepest gratitude for the reviewer’s time and constructive comments. It is our pleasure to address the primary concerns in the following sections in addition to the general rebuttal given above.
>
> **Interpretability advantage over black box+grad-cam**
> Compared to Grad-CAM, although we could readily offer more detailed visualizations (e.g., multiple heatmaps for prototypical parts of each class instead of only a single heatmap for each class as in Grad-CAM), we argue that *the visualization is neither our only advantage, nor the most important one*. A much more important advantage is: Grad-CAM only tells what the network is looking at, but cannot tell what prototypical parts the focused area looks similar to, lacking the explanation on how these areas are actually used for decision [1]. In contrast, our method clearly tells what prototype the focused area looks similar to, how similar they are, and how important is that prototype in classifying an image to a certain class. Thus our method offers a much more transparent reasoning process by design. An example is given in section 4, last paragraph. Since Grad-CAM does not even have these capabilities, it's hard to show prototype based models are quantitatively more interpretable (section 1, paragraph 5 of our baseline [1] also claims this). Regarding the experiment on the potential usage for test-time intervention as a benefit: thanks for your suggestion! We add a case study in section A.4 of the appendix using re-parameterized NMF prototypes. This section demonstrates how human could intervene by setting the importance of the intervened prototype to zero using expert knowledge to correct the model's prediction. We show that assuming an expert can intervene one most important prototype once when correcting the prediction, the Top1 performance of coc-tiny can increase from $71.9\%$ to $82.7\%$ in ImageNet.
>
> [1] Chen, Chaofan, et al. "This looks like that: deep learning for interpretable image recognition." Advances in neural information processing systems 32 (2019).
>
> **Writing and presentation**
> Thank you for pointing this out! We have improved the method description in the revised manuscript. Regarding the diversity of prototypes: Note that we do not claim our prototypes to be more diverse than ProtopNet and we acknowledge that the perceptual diversity is hard to measure because the similarity of prototypes in the latent space doesn't necessarily match the human perceptual similarity [2]. Instead, we claim that they are more comprehensive (diverse yet complete, as explained in page 2, line 4) and offer quantitative results in Table 1. Seeking for pure diversity might not be that meaningful because even a set of random prototypes could be very diverse.
>
> [2] Nauta, Meike, Ron Van Bree, and Christin Seifert. "Neural prototype trees for interpretable fine-grained image recognition." Proceedings of the IEEE/CVF Conference on Computer Vision and Pattern Recognition. 2021.
>
>
> **The standard ResNet34** The result is already in the existing table. Our method only tries to decompose the classification head of a trained model for more transparent prototype based reasoning process, and thus has no influence on the performance.
>
> **Results in Table 3**
> We refer to the common response to all reviewers, where a simple re-parameterization could remove the not-so-interpretable residual prototype and increase the discriminative power of NMF prototypes, yet maintaining the interpretability of these prototypes.
>
> We hope our rebuttal could address your concerns and we sincerely hope that the reviewer could consider increasing the score.

---

### Author Response · Authors · 2023-11-20
**General response (part I)**

**General response** We would like to thank all reviewers for taking their time to review and provide constructive feedback on our work. The comments are very thorough and provide insights into how we can improve the paper. We are highly motivated to see that reviewers think the idea of turning a black box model into a more interpretable one is interesting [Reviewer eeCj], the proposed method have good interpretability without losing the performance [Reviewer A726, XoPR] and comprehensive evaluations of different architectures as well as various aspects of the method are offered [Reviewer eeCj, XoPR]. We are very encouraged by your posititve feedback and look forward to further engagements in the discussion!

**Common concern regarding the performance of NMF prototypes [Reviewer eeCj, A726, XoPR]** we suggest that **a simple re-parameterization can increase the discriminativeness of the NMF prototypes and remove the residual prototype without sacrificing the interpretability**. We describe how to re-parameterize, and offer both theoretical analysis and empirical evidence to show that the interpretability is not sacrificed in following paragraphs.

**How to re-parameterize** Formally, we denote the classification head of class $c$ as $V^c\in \mathbb{R}^{1\times D}$. Before the re-parameterization, it is decomposed into NMF prototypes and a residual prototype as follows:
\begin{equation}
    V^c = a_1b_1+a_2b_2+...+a_pb_p+R^c
\end{equation}
where $a_i$ are the scalers in $C^c\in \mathbb{R}^{1\times p}$ indicating the importance of each prototype [main paper equation 4], $b_i\in \mathbb{R}^{1\times D}$ indicates the $i^{th}$ NMF prototype obtained in $B^c\in \mathbb{R}^{p\times D}$ [main paper equation 1], and $R^c\in \mathbb{R}^{1\times D}$ is the residual prototype [main paper equation 5]. Since the residual prototype might be less interpretable, we point out that the following simple strategy can easily distribute the residual prototype's parameters to NMF prototypes and remove the explicit inference dependence on it. The new inference process is based on $p$ augmented NMF prototypes:
\begin{equation}
    V^c = a_1(b_1+\frac{R^c}{\sum_{i=1}^p a_i})+...+a_p(b_p+\frac{R^c}{\sum_{i=1}^p a_i})
\end{equation}
To this end, we successfully obtain the NMF prototypes in the form of $b_i +\frac{R^c}{\sum_{i=1}^p a_i}$ with enhanced discriminative power and remove the explicit residual prototype $R^c$ in the inference.

**Interpretability visualization** of the augmented NMF prototypes is created via $E^{c'}$ of the following convex optimization, which is the same form as original NMF prototypes's visualization:
\begin{equation}
    \min\limits _{E^{c'}} ||A^c - E^{c'}B^{c'}||_2^2
\end{equation}

where $A^c\in \mathbb{R}^{nHW\times D}$ indicates the extracted features, the rows of $B^{c'}\in \mathbb{R}^{p\times D}$ are the augmented NMF prototypes $b_i +\frac{R^c}{\sum_{i=1}^p a_i}$, and $E^{c'} \in \mathbb{R}^{nHW\times p}$ is the new encoding matrix for visualization, indicating the strength of $p$ prototypes in each position of $n$ images with height $H$ and width $W$. We refer to Fig. 7 of the appendix to show empirical evidence that the prototypes with enhanced discriminative power still deliver interpretable parts based representations. In architectures with inductive bias such as ResNet34 and CoC-tiny, the re-parameterized prototypes in the foreground become even better separated from the background prototype, compare blue/green areas with red areas.

---

> ### Author Response · Authors · 2023-11-20
> **General response (part II)**
>
> **Why this re-parameterization does not sacrifice the interpretability**  We focus the analysis on the difference between encoding matrix $E^c$ and $E^{c'}$ for interpretability visualization. Observe the equation for re-parameterization, we are actually shifting every axis of the coordinate system consisting of $p$ prototypes by the vector $\frac{R^c}{\sum_{i=1}^p a_i}$. Therefore, the change in the interpretability visualization is reflected by the change of the coordinates of all features in this coordinate system. For the ease of analysis, first assume each prototype is orthogonal to each other. Consider any prototype $b_i$, the strength of the $m^{th}$ feature patch $A^c_m$ from class c along the $k^{th}$ prototype $b_k$ is $A^c_m b_k^T$ before re-parameterization. After the re-parameterization, the projected strength becomes $A^c_m b_k^T + A^c_m (\frac{R^c}{\sum_{i=1}^p a_i})^T$. Therefore, the change of the visualization completely depends on the term $A^c_m (\frac{R^c}{\sum_{i=1}^p a_i})^T$. Fortunately, it's reasonable to expect this term to be very small compared to the first term $A^c_m b_k^T$ especially in the salient areas where the first term is large and contribute the most to the interpretability visualization. This is because original NMF prototypes $B^c$ are explicitly optimized to approximate the features $A^c$. Though we use the assumption that prototypes are strictly orthogonal to derive this conclusion, a similar logic also applies to not strictly orthogonal NMF prototypes, because NMF is exactly an optimization process to make the prototypes as diverse/orthogonal as possible, yet serving as a good basis to span a feature space covering the features of the target class.
>
> We have accordingly revised the section 3.3 and 4.2 of the manuscript and A.2, A.3, A.4 of the appendix (after the reference section). Modified parts are colored blue.

---

### Meta-Review · Area_Chair_8Pif · 2023-12-10

**Metareview:**

This paper proposes ProtoNMF, which can turn a black-box model into a prototype-based interpretable model using non-negative matrix factorization (NMF). Given a feature matrix for a specific class, the authors apply NMF to it to factorize it into the prototype basis vectors and the coefficient vectors that represent the coordinates of the image features along the prototype basis vectors, and then introduce a residual prototype to prevent the loss of accuracy. The authors demonstrated the method's efficacy on two benchmark datasets.

Although all reviewers seem to find the proposed method of explaining black-box models as promising, the paper received unanimously negative reviews , due to the following concerns:
- The lack of discriminativeness of the generated explanations which renders the effectiveness of the proposed method questionable.
- The loss of accuracy when not using the residual prototypes.
- The lack of clarity in writing, such as in the inference procedure, description of the residual prototypes, and linear combination of the prototypes.

To address the concerns, the authors made a complete revision of the paper and proposed a new reparameterization that does not require residual prototypes, which resulted in some of the reviewers adjusting their scores. Yet, all reviewers maintained their negative ratings as the clarity and presentation should be further enhanced. Thus, the paper would benefit from another round of reviewing process, especially considering that the authors proposed a completely revamped method with a new reparameterization.

**Justification For Why Not Higher Score:**

The proposed method had some limitations that should have been addressed, and thus its effectiveness in terms of interpretability was questionable. Although they were addressed by the authors during the rebuttal period with the introduction of a new reparameterization, because this is a new method it should be evaluated in another reviewing round. The overall clarity was improved in the revised version of the paper but the reviewers find that it needs more work.

**Justification For Why Not Lower Score:**

N/A

---

### Decision · Program_Chairs · 2024-01-16

Reject